# The evolutionary and molecular history of a chikungunya virus outbreak lineage

**Janina Krambrich**[1‡]**, Filip Mihalič**[1]**, Michael W. Gaunt**[2]**, Jon Bohlin**[3]**, Jenny C. Hesson**[1,4‡]**, Åke Lundkvist**[1‡]**, Xavier de Lamballerie**[5]**, Cixiu Li**[6,7]**, Weifeng Shi**[7,8]**, John H.-O. Pettersson** [9,10,11,12‡] *

**1** Department of Medical Biochemistry and Microbiology, Uppsala University, Uppsala, Sweden, **2** Solena Ag, Foster City, California, United States of America, **3** Infectious Disease Control and Environmental Health, Norwegian Institute of Public Health, Oslo, Norway, **4** Biologisk Myggkontroll, Nedre Dalälvens Utvecklings AB, Gysinge, Sweden, **5** Unité des Virus Émergents (UVE), Aix-Marseille University—IRD 190—Inserm 1207, Marseille, France, **6** Key Laboratory of Emerging Infectious Diseases in Universities of Shandong, Shandong First Medical University & Shandong Academy of Medical Sciences, Taian, China, **7** Department of Infectious Diseases, Ruijin Hospital, Shanghai Jiao Tong University School of Medicine, Shanghai, China, **8** Shanghai Institute of Virology, Shanghai Jiao Tong University School of Medicine, Shanghai, China, **9** Department of Medical Science, Uppsala University Uppsala, Sweden, **10** Department of Clinical Microbiology and Hospital Hygiene, Uppsala University Hospital, Uppsala, Sweden, **11** Department of Microbiology, Public Health Agency of Sweden, Solna, Sweden, **12** Department of Microbiology and Immunology, Peter Doherty Institute for Infection and Immunity, University of Melbourne, Melbourne, Victoria, Australia

‡ Members of the Zoonosis Science Center, Uppsala University, Sweden
* john.pettersson@uu.se, john.pettersson@folkhalsomyndigheten.se (JHOP)

**Data Availability Statement:** All CHIKV sequences were deposited to the NCBI GenBank (https://www.ncbi.nlm.nih.gov/genbank/) under accession

## Abstract

In 2018–2019, Thailand experienced a nationwide spread of chikungunya virus (CHIKV), with approximately 15,000 confirmed cases of disease reported. Here, we investigated the evolutionary and molecular history of the East/Central/South African (ECSA) genotype to determine the origins of the 2018–2019 CHIKV outbreak in Thailand. This was done using newly sequenced clinical samples from travellers returning to Sweden from Thailand in late 2018 and early 2019 and previously published genome sequences. Our phylogeographic analysis showed that before the outbreak in Thailand, the Indian Ocean lineage (IOL) found within the ESCA, had evolved and circulated in East Africa, South Asia, and Southeast Asia for about 15 years. In the first half of 2017, an introduction occurred into Thailand from another South Asian country, most likely Bangladesh, which subsequently developed into a large outbreak in Thailand with export to neighbouring countries. Based on comparative phylogenetic analyses of the complete CHIKV genome and protein modelling, we identified several mutations in the E1/E2 spike complex, such as E1 K211E and E2 V264A, which are highly relevant as they may lead to changes in vector competence, transmission efficiency and pathogenicity of the virus. A number of mutations (E2 G205S, Nsp3 D372E, Nsp2 V793A), that emerged shortly before the outbreak of the virus in Thailand in 2018 may have altered antibody binding and recognition due to their position. This study not only improves our understanding of the factors contributing to the epidemic in Southeast Asia, but also has implications for the development of effective response strategies and the potential development of new vaccines.

numbers PP193832–PP193843 and the raw data (excluding human reads) was deposited under NCBI SRA (https://www.ncbi.nlm.nih.gov/sra) accession nr: PRJNA1066385.

**Funding:** JHOP is funded by the Swedish Research Council Vetenskapsrådet (grant no.: 2020-02593). This study was partially funded by the Academic Promotion Programme of Shandong First Medical University (grant no.: 2019QL006). JK was funded by the European Union's Horizon 2020 research innovation program (grant no.: 874735 (VEO)), and the SciLifeLab Pandemic Preparedness projects (grant no.: LPP1-007 and REPLP1:005). The funders had no role in study design, data collection and analysis, decision to publish, or preparation of the manuscript.

**Competing interests:** The authors have declared that no competing interests exist.

## Author summary

We investigated the evolutionary and molecular history of the East/Central/South African (ECSA) genotype to determine the origins of the 2018–2019 chikungunya virus (CHIKV) outbreak in Thailand. We used newly sequenced clinical samples from travellers returning to Sweden from Thailand in late 2018 and early 2019 together with previously published genome sequences. Our phylogeographic analysis shows that the Indian Ocean lineage (IOL), found within ECSA, evolved in Eastern Africa, Southern Asia, and Southeast Asia for about 15 years before the outbreak in Thailand in 2018. We have also identified amino acid substitutions that may be associated with immune evasion, increased spread, and higher virulence that occurred prior to the outbreak and may have played a critical role in the rapid spread of the virus. Our study concludes that monitoring and understanding CHIKV dynamics remains critical for an effective response to the previously unpredictable outbreaks of the virus.

## Introduction

Chikungunya virus (CHIKV, *Togaviridae*), is a single-strand positive-sense mosquito-borne RNA virus with a genome of approximately 12 kb that comprises two open reading frames (ORFs) encoding non-structural and structural proteins respectively [1]. The virus is transmitted to humans mainly through the bites of infected mosquitoes, such as *Aedes aegypti* and *Aedes albopictus*, which are widely distributed in tropical and subtropical regions around the world [2–5]. These mosquito species are also responsible for the transmission of other well-known viruses, for example dengue virus, Zika virus, and yellow fever virus [2]. Since the discovery of CHIKV in Tanzania in 1952, the virus has been identified as the causative agent of multiple outbreaks occurring globally and suspected to be the pathogen of outbreaks dating back centuries [6,7]. However, since 2004 the virus has spread rapidly to new geographic regions and cases are now reported from over 100 countries in Asia, Africa, Europe, and the Americas [8,9]. The geographic distribution of CHIKV is primarily determined by the presence and spread of its mosquito vectors.

In African forests, a sylvatic cycle of CHIKV occurs between mosquitoes and non-human primates [10,11]. This sylvatic cycle may lead to sporadic spill-over events, where the virus is transmitted to humans, initiating a separate urban cycle [12]. In the urban cycle, non-human primates are not necessary to sustain the epidemic, since the virus is transmitted between humans and *Aedes* mosquitoes. The sylvatic and the urban cycles can exist separately, contributing to the complex transmission dynamics of CHIKV in African regions.

Following the expansion of CHIKV since 2004, outbreaks have occurred throughout the tropical- and subtropical regions of the world, becoming a significant public health concern. Between 2004 and 2020, 3.4 million suspected and confirmed CHIKV cases were reported from various countries [13]. The actual number of CHIKV infections is likely considerably higher due to underreporting and asymptomatic cases. CHIKV infections, although rarely fatal, can lead to prolonged and incapacitating joint pain, lasting months or even years in some cases [9,14,15]. There are several vaccines for CHIKV that are currently under development. The Coalition for Epidemic Preparedness Innovations and the European Commission are currently supporting the development of a live-attenuated, single-dose vaccine that is designed by deleting a part of the CHIKV genome (Ixchiq, VLA1553 by Valneva). In November 2023, the US Food and Drug Administration approved and authorized this vaccine in the US [16,17].

The Jenner Institute research group has developed another CHIKV vaccine using a combination of recombinant chimpanzee adenoviruses and Modified vaccinia Ankara (MVA), which however is not approved for use yet [18]. Other prevention efforts focus primarily on reducing mosquito populations and avoiding mosquito bites. As of now, there is no specific treatment for CHIKV [19].

Based on phylogenetic analyses, CHIKV is commonly divided into three major lineages: the East/Central/South African (ECSA), the West African, and the Asian lineages [20,21]. The ECSA lineage gave rise to the Indian Ocean lineage (IOL), which has been responsible for epidemics in the Indian Ocean islands, South and Southeast Asia, and Europe since 2005 [21,22]. The first CHIKV outbreak in Thailand was reported in Bangkok in 1958, and the Asian genotype was identified as the cause of that outbreak [23]. The next notable outbreak occurred in southern Thailand between 2008 and 2009, followed by a smaller local spread in 2013 in north-eastern Thailand, both caused by the ECSA genotype [24,25]. The overall number of reported cases remained low until just before the start of the 2018–2019 outbreak, according to the Bureau of Epidemiology in Thailand. In June 2018, the number of monthly reported chikungunya cases in Thailand began to increase and a nationwide spread of CHIKV was observed with approximately 15,000 confirmed cases reported between 2018 and 2019 [26]. The virus primarily affected urban and semi-urban areas, with high transmission rates observed in densely populated regions. Due to international travel and the popularity of Thailand and other tropical regions as tourist destinations, an increase in imported CHIKV cases to other countries, including Europe and the United States, was observed both after the outbreak in Thailand in 2018 and after the CHIKV outbreak in the Caribbean and South America in 2014 [27–32].

To improve our understanding of the factors that contributed to this epidemic in Southeast Asia, and particularly the sudden increase of cases in the 2018 Thailand outbreak, we conducted a comprehensive phylogeographic and outbreak lineage analysis focusing on the ECSA and IOL. We reconstructed the evolutionary and geographic history, divergence times and performed mutational profiling. We also carried out protein folding predictions from genes related to transmission and virulence. The analyses were performed on previously sequenced genomes as well as newly sequenced clinical samples from travellers who returned to Sweden from Thailand in late 2018 and early 2019.

## Methods

### Ethical statement

This study was in part conducted at the Public Health Agency of Sweden supported by the ordinance (2021:248:§37) from the Swedish Parliament to study and monitor the situation and development of infectious diseases. It should be noted that, apart from the country of infection and the date of sample collection, no information or data from this project can be linked or traced to a specific individual include in the study. Therefore, the CHIKV-positive samples were used in accordance with the regulations governing the use of such material and in accordance with the mandate of the Swedish Parliament.

### Preparation of patient material

Serum samples from a total of 12 patients who had travelled from Thailand to Sweden between December 2018 and April 2019 and who were PCR-positive for CHIKV in a real-time PCR screening were included in the study (S1 Table). Total RNA was extracted from all patient samples by automated magnetic bead total nucleic acid extraction using a MagLEAD system (Precision System Science Co.) from the aqueous phase after Trizol–chloroform separation.

RNA-seq libraries were then prepared using the Trio RNA-Seq Library Preparation Kit (NuGen) according to the manufacturer's instructions and subsequently sequenced on one Illumina X10 lane. RNAseq library preparation and high-throughput sequencing were performed by BGI, Hong Kong.

## Sequence processing

First, low-quality reads were removed with Trimmomatic v.0.36 [33] employing the default settings (i.e., 'ILLUMINACLIP:TruSeq3-PE.fa:2:30:10:2:True LEADING:3 TRAILING:3 MIN-LEN:36'). All quality-checked sequence data libraries were then mapped with Bowtie v.2.3.4. using the default local settings (i.e.,—sensitive-local, same as: -D 15 -R 2 -N 0 -L 20 -i S,1,0.75) against a NCBI CHIKV reference sequence (NCBI GenBank accession number: MF773566), whereupon 50% majority consensus sequences were generated. All CHIKV sequences were deposited to the NCBI GenBank (https://www.ncbi.nlm.nih.gov/genbank/) under accession numbers PP193832–PP193843 and the raw data (excluding human reads) was deposited under NCBI SRA (https://www.ncbi.nlm.nih.gov/sra) accession nr: PRJNA1066385.

## Evolutionary analyses

First, 2,564 CHIKV sequences of ≥8000bp, with known collection date and geographic location were retrieved from NCBI GenBank and aligned together with the 12 CHIKV consensus genome sequences generated above using MAFFT v.7.520 [34,35] utilizing the L-INS-i algorithm, where the 5' and 3' ends were trimmed. To reduce the number of sequences prior to temporal analyses, we constructed a maximum likelihood phylogenetic tree with IQ-TREE v.2.2.0 [36] using the Generalized time-reversible model of Tavaré 1986 (GTR) with empirical base frequencies, invariant sites, and invariant sites plus FreeRate model with tree categories (GTR+F+I+I+R3) following the ModelFinder implemented in IQ-TREE [37]. We then subsampled the phylogenetic tree to include a total of 218 CHIKV ECSA genotype sequences (see coloured terminal nodes in S1 Fig for sequences included), which were used for the temporal and evolutionary analyses. The temporal structure of the subsampled dataset, sampled between the years 1953 and 2023, was then assessed using TempEst v.1.5.3 (S2 Fig) [38]. Finally, the evolutionary history of the subsampled dataset was assessed using BEAST v.1.10.4 [39,40] by performing a single run of 250 million MCMC generations, sampling every 5k generations, using terminal node calendar dates (i.e. tip dates) as temporal calibration, GTR with invariant sites and four gamma variables with default flat Dirichlet priors as a model of nucleotide evolution, an uncorrelated lognormal relaxed molecular clock with default prior distribution, and a non-parametric Gaussian Markov random field Bayesian Skyride tree prior [41]. Following 10% burn-in, the run was checked using Tracer v.1.7.2 [42] to confirm that the effective sample size for all parameters was >200. Finally, we used TreeAnnotator v.1.10.4 [40] to compute a maximum clade credibility tree and calculate median node heights. The resulting tree was viewed and annotated in FigTree v.1.4.4 [43]. Posterior probabilities ≥ 0.95 for key nodes in Fig 1 are presented in S3 Fig.

The presented analysis is based on a subset of sequences. While we aimed to select a representative subset with minimal bias, it may not capture the full diversity of the population, potentially leading to biased results. Important evolutionary information in sequences not included may be missed, affecting the accuracy of our findings. Additionally, rare variants or intermediate forms may not be included, limiting the resolution of evolutionary pathways. This could result in an artificially homogeneous subset, underestimating true diversity and evolutionary pressures.

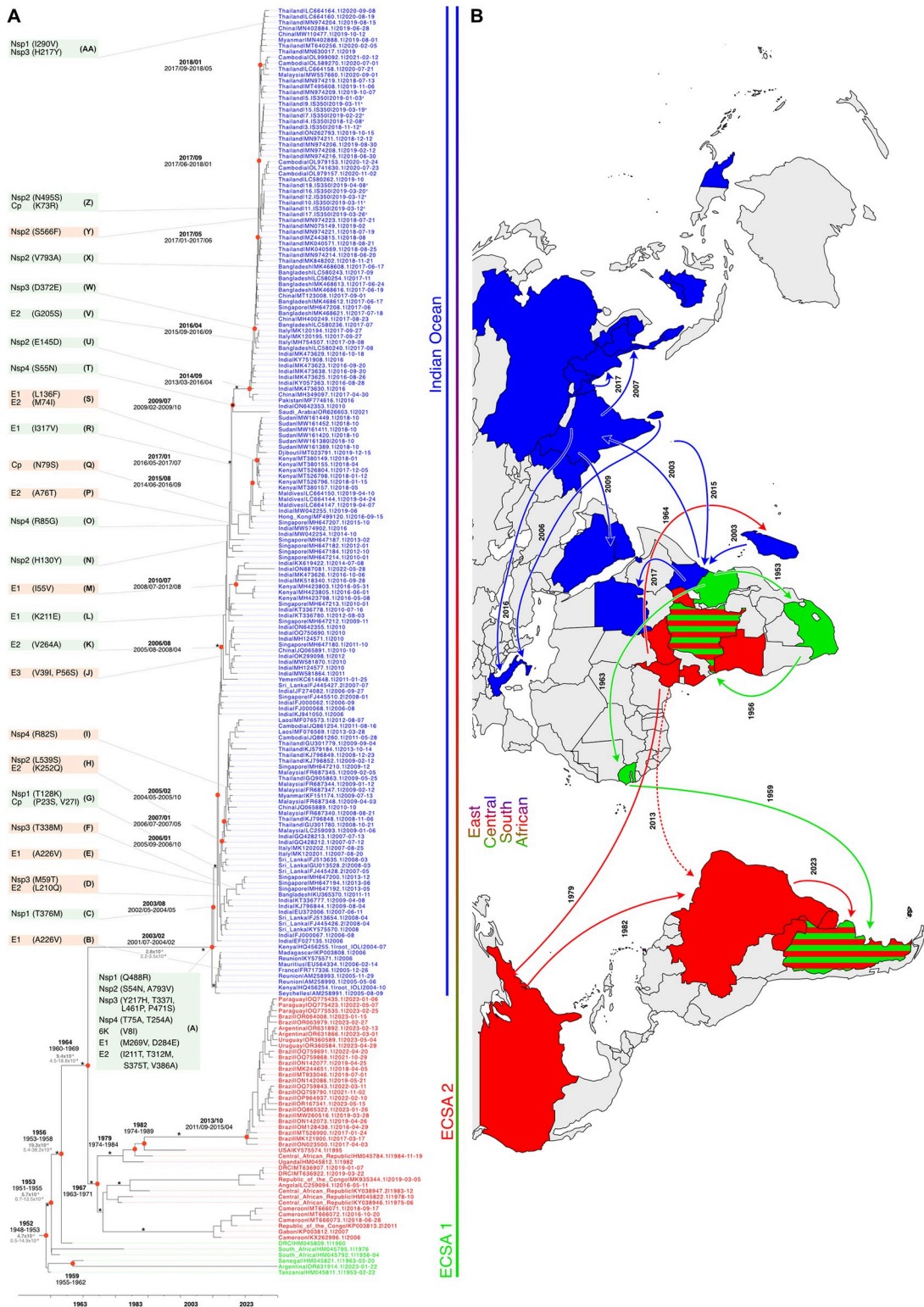

**Fig 1. Phylogenetic and Phylogeographic Analysis of the ECSA CHIKV samples included in the present study.** Phylogenetic tree of the CHIKV ECSA genotype. Strains are colour-coded by CHIKV lineage (where light green represents ECSA 1, red ECSA 2, and blue IOL). Strains sequenced in this study are indicated by an ˣ. Black values at red circles indicate branching times, grey values display the mutational speed. Major (basal) branches with posterior probabilities of ≥0.95 are indicated by an *. Amino acid substitutions at different alphabetically characterized nodes are indicated in green boxes if affecting the entire IOL and in

orange boxes if affecting only certain clades. Countries are colour-coded according to the phylogenetic tree, and suggested transmission events are shown with arrows on the global map. Uninterrupted lines on the map show data from our investigation, while dashed lines display previous assumptions of spread. The base layer of the map was retrieved from Wikimedia Commons and modified under a Creative Commons Attribution 4.0 International license (https://commons.wikimedia.org/wiki/File:Blank_World_Map_%28in_the_year_2024_and_with_borders_and_blue_oceans%29.png).

## Mutational analysis

In order to ascribe potential change in protein function due to emergent mutations arising in the IOL, we analysed the location of the proposed mutations by determining their physical location on the existing structural models of chikungunya protein complexes obtained from Protein Data Bank (PDB). This approach was used to visualise mutations in the proteins of the replication and spike complexes respectively. To gain insight into mutations that were located outside of the experimentally resolved regions we additionally predicted the structural models using ColabFold [44]. ColabFold predictions were in excellent agreement with the solved crystal structures relative to globular domains and also allowed us to visualise the disordered regions found in viral proteins. The replication complex mutations were mapped on the structure accessible under PDBid entry 7y38 (complex of Nsp1, Nsp2 helicase domain and Nsp4) [45] and 4ztb (Nsp2 protease domain) [46]. It should be noted that not all of the amino-acid residues are resolved in these crystal structures. For the E1-E2-E3 trimer spike complex we visualised mutations based on the PDBid entry 6jo8 [47] and 6nk6 [47]. Finally, for the Nsp3, 6K and CP we mapped the mutations directly onto the ColabFold structure prediction due to the high degree of disorder (Nsp3, CP) or lack of other structural information (6K). For both Nsp3 and CP, we superimposed ColabFold predictions onto the relevant solved crystal structures of their respective globular domains to confirm the quality of the predictions and found them to be in excellent agreement (Nsp3 Macrodomain, PDBid: 6vuq, RMSD: 0.344; Nsp3 Zinc-binding domain, PDBid: 4gua, RMSD: 0.58; CP protease domain, PDBid: 5h23, RMSD: 0.505). PyMOL (The PyMOL Molecular Graphics System, Version 2.5.4 Schrödinger, LLC) was used to visualise the protein structures. Different protein sequences were used for the experimentally determined structures meaning that the stick model of the mutation used for figures does not always corresponds to the amino acids involved in the mutation we discuss in text (example: for the Nsp4 structure in Fig 2D the T75A mutation, a methionine (M) is the actual residue in the Nsp4 sequence that was used for Cryo-electron microscopy experiment, and is thus visualised in the figures). With that in mind the highlighted residues are intended to indicate the position of the mutation rather than imply any amino-acid change or impact of the mutation itself. The 3D models from X-ray crystallography offer only a static snapshot of protein structures. Since most protein complexes, including the spike protein, are dynamic and undergo significant structural changes (e.g., during membrane fusion), our analysis will miss the impact of mutations on the conformations that have not been structurally solved. Additionally, interpretation of mutation effects on dynamic properties of proteins is not possible. Finally, the predicted structures from ColabFold, though of high quality (indicated by high pLDDT scores), are still predictions and not experimentally verified. Therefore, their interpretations should be taken with caution.

## Results

### Evolutionary history of CHIKV

Based on the phylogenetic analysis, CHIKV can be divided into three different genotypes (S1 Fig), the West African, the Asian, and the ECSA genotype, which are supported by high posterior probabilities (S3 Fig). High posterior probabilities ($\geq 0.95$) were also observed for all

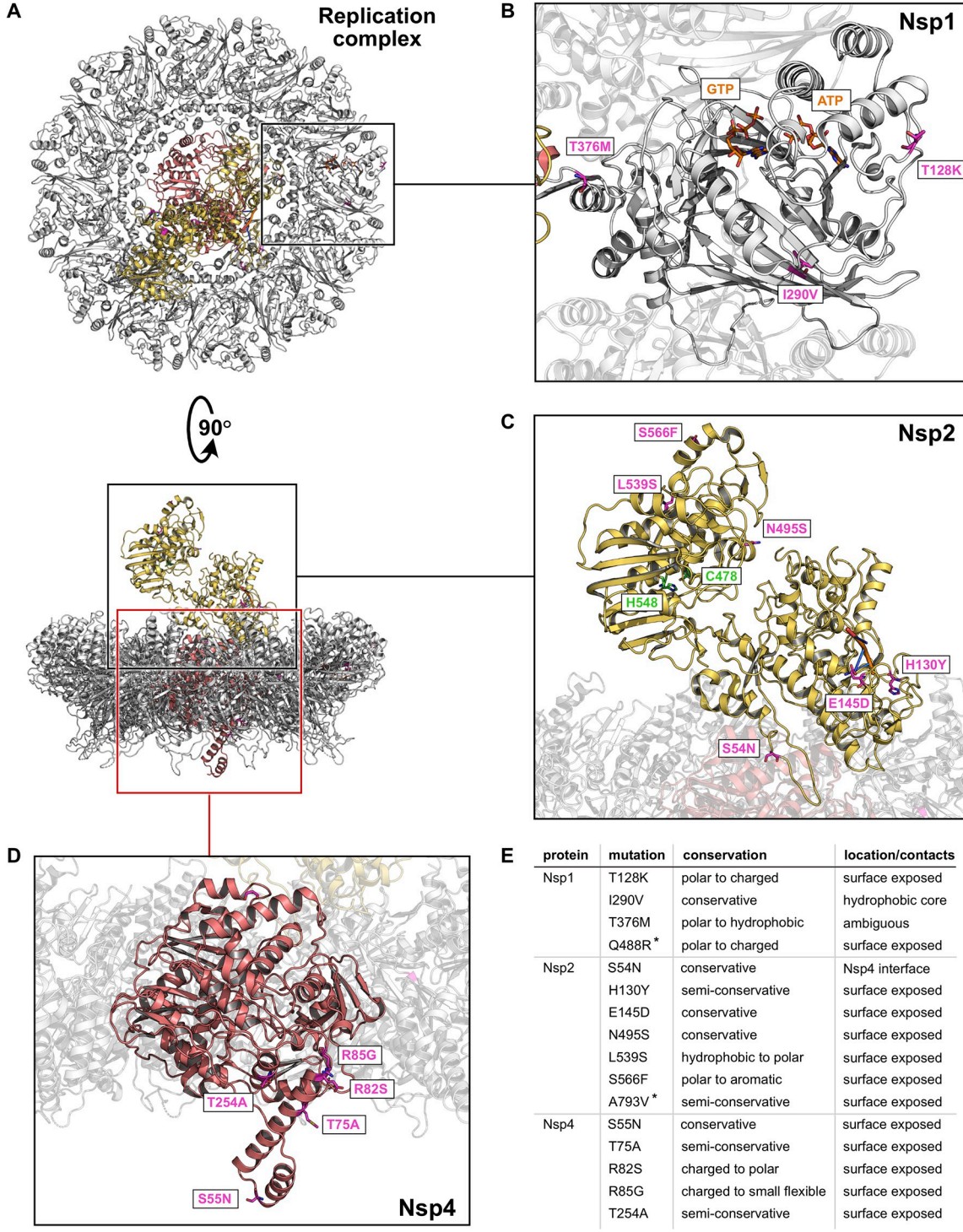

**Fig 2. Emergent mutations in the Nsp1-Nsp2-Nsp4 replication complex. A)** Representative view of the CHIKV replication complex. Nsp1 is in grey, Nsp2 in yellow and Nsp4 in pink colour. Structural models for visualization were obtained from PDBid 7y38 [45] and 4ztb [46]. In all panels the mutations arising in the IOL are shown as purple sticks. Note that highlighted amino acids correspond to the position of the indicated mutations but not always to the actual amino acid involved in the mutation process. (see methods for more information) **B)** Close-up view, highlighting the mutations on Nsp1. The GTP and ATP which are cofactors for the Nsp1 are also shown as orange sticks. **C)** Model of Nsp2 with indicated mutations. The catalytic residues are shown as green sticks and the RNA fragment bound to the helicase domain is shown as orange sticks. **D)** Detailed view showcasing the mutations on Nsp4. **E)** Summary of all mutations found in the replication complex.

nodes, which are described in the results section. The ECSA genotype is characterised by three clusters, as shown in Fig 1: the paraphyletic group ECSA 1 (light green) and the monophyletic groups ECSA 2 (red) and IOL (blue) (see also S1 Fig). When analysing the complete ECSA genotype (consisting of light green, red, and blue, see S2 Fig), the root-to-tip analysis showed that our investigated data set exhibited a significant temporal structure (correlation coefficient = 0.90; R-squared = 0.82, p<0.001, S2 Fig). To investigate the dispersal history and temporal structure of the CHIKV ECSA genotype and the IOL in particular, we then performed a phylogeographic analysis of 218 viral genomes of this genotype, covering approximately 70 years of CHIKV evolution and movement between regions (Fig 1). Fig 1 shows the resulting time-calibrated phylogenetic tree with maximum clade credibility of CHIKV together with a map indicating the major routes of CHIKV movement for ECSA lineages 1 and 2 (green and red) and the IOL (blue).

For the ECSA genotype as a whole, i.e., including the IOL, the lognormal, uncorrelated relaxed clock estimated the mean evolutionary rate to be 4.4 x $10^{-4}$ substitutions/site/year (95% highest posterior density [95% HPD] = 4.1–4.8 x $10^{-4}$) with a most recent common ancestor emerging around 1952 [95% HPD = 1948–1953] (Fig 1). For the IOL, the mean evolutionary rate was estimated to be 2.8 x $10^{-4}$ substitutions/site/year (95% HPD = 2.2–3.5 x $10^{-4}$) and a most recent common ancestor emerging around February 2003 (95% HPD 2001–07 to 2004–02). However, we recognise, that there are rate variations between lineages even within the ECSA genotype [48], affecting the African clades, which have a lower sampling frequency. It is noteworthy that the earliest recorded CHIKV isolates are from West and East Africa, emphasising the historical importance of this region in the evolutionary trajectory of the virus.

## Early evolution and movement of the ECSA genotype

Following the divergence from a West African ancestor (S1 Fig) [48], our phylogeographic analysis of the ECSA genotype reveals a highly structured dispersal network with a significant interregional spread of CHIKV. Since the emergence of the most recent common ancestor of the ESCA genotype, CHIKV has circulated to date, moving between many sub-Saharan African countries, particularly in eastern, central, and southern African countries. The first CHIKV discoveries and isolations were limited to East Africa in 1953 (Tanzania) and West Africa in 1964 (Nigeria) (Figs 1 and S1). Between the 1950s and the 1980s, African ECSA-- CHIKV viruses continued to diversify, followed by a long period of sporadic detection but possibly continuous sylvatic circulation [10]. The first of the evolved ECSA lineages, African ECSA 1 (light green), showed intermediate-range dissemination within sub-Saharan Africa, spreading from Central Africa to the Americas (red) in the late 1980s and early 2010s (1995 in the USA and 2014 in Brazil) (Fig 1). Subsequently, the African ECSA 1 lineage spread eastwards to the Indian Ocean islands (Madagascar, Mauritius, Mayotte, Comoros) in the 1960s and to India in the 21$^{st}$ century (blue) (Fig 1). A more recent African cluster with representatives from Cameroon, Gabon and the Democratic Republic of Congo, probably emerged more than 20 years ago and circulated in this region until at least 2018. A similar but independent cluster in Angola and the Democratic Republic of Congo, was separated by more than 30 years of evolution before it emerged in 2011. A comparable scenario occurred when the pathogen was introduced to Brazil via Western Africa or the USA after circa 30 years of unaccounted evolution, which later led to a significant and sustained outbreak in South America [49]. The uncertainty if the virus was introduced to Brazil directly from the African continent or via the US arises from our inability to verify whether the sequence USA|KY575574.1|1995 pertains to a mosquito collected in the US or a human infected locally in the US rather than a person

infected in an African country returning to the US (i.e., an imported case to the US). Our results suggest an introduction to Brazil via the US. This is depicted by the continuous line from Africa to the US in 1979 and a second line down to Brazil in 1982 in Fig 1. The uncertainty is depicted by adding a broken line with the second option of a direct introduction to Brazil from Africa in Fig 1. The Southeast Asian region in particular, including India, emerged as a major inter-regional transmission hub, facilitating the spread of CHIKV to other regions. Sub-Saharan Africa, where the virus originally emerged in the 1920s, also played a central role in the global spread and maintenance of the virus, albeit to a lesser extent than Southeast Asia and India. Bidirectional transmission events between Africa and Asia have been documented, including early transmission from Africa to India in the late 1920s, with subsequent introductions in 1986 and 2000.

## Spread and recent emergence of the IOL in Southeast Asia

The emergence and spread of CHIKV in the Indian Ocean islands, the Indian subcontinent, and Southeast Asia are associated with a significant increase in cases. Here we examine the evolutionary history of CHIKV to trace the origins of the outbreak lineage that caused a substantial number of cases in Thailand and other Southeast Asian countries in 2018 and 2019 [24,26]. The 2004 outbreak of CHIKV in the Indian Ocean islands was the first documented outbreak in the IOL [50]. The outbreak was mainly observed in urban and semi-urban areas, for example on the Comoros Islands where more than 5,000 cases were reported [51]. Seroprevalence studies from 2011 indicate that 20% of the population on Ngazidja (Grande Comore), the largest island in the Comoros with a population of approximately 316,600, were infected with CHIKV [52]. The outbreak then spread to other islands in the Indian Ocean, including Madagascar, Mauritius, and the Seychelles, and eventually to other parts of the world, including Europe [53,54]. The IOL has been circulating in South and Southeast Asia for two decades now, with several sub-lineages and variants having emerged and spread throughout the region (Fig 1). The last common ancestor of the IOL of the ECSA genotype is estimated to have originated in coastal Kenya and the Mascarene islands, around early 2003 (95% HPD: 2001–07–2004–02), which is consistent with previous estimates [50,55]. A new IOL sub-lineage, distinct from the previous IOL that originated from the Kenyan coast, was found to have originated in India and circulated during 2008–2016, with subsequent spread to Pakistan, Bangladesh, Thailand, and Italy [56]. Several introductions and re-introductions of IOL strains to Africa (Kenya, Djibouti, and Sudan) and the Arabian Peninsula have been observed over the years. Most outbreaks in South Asian countries since 2005 have reportedly been caused by IOL strains, and new clades have evolved in multiple Southeast Asian countries over time, indicating a significant presence of the IOL in the region.

## Emergence of mutations in the IOL preceding the Thai outbreak in 2018

The CHIKV sequences isolated in this study from Swedish travellers, as well as other sequences isolated during the 2018–2019 outbreak in Thailand, belonged to the IOL but show marked differences from the strains responsible for the massive CHIKV outbreak in Thailand in 2008–2009, indicating a clear, non-local origin. The outbreak was likely due to the introduction of a viral strain from South Asia, possibly Bangladesh (Fig 1) [56], since phylogenetic analysis of the isolates revealed that the Thai sequences diverged from a Bangladeshi ancestor around April 2017 (95% HPD: January–May 2017). Following its introduction and epidemic spread in Thailand, CHIKV also spread to Cambodia, Malaysia, Myanmar and China starting in mid-2018 (Fig 1). Our analysis of the IOL outbreak strains led to the detection of mutations in the ancestral strain, distinguishing the outbreak strains from ESCA 1 and 2 at the last common

ancestor node (Fig 1, Node A). Our results revealed a high frequency of amino acid substitutions in both structural and non-structural genes of CHIKV. Nine substitutions were detected in non-structural proteins and nine in structural proteins. Over the subsequent history of the IOL, 18 additional amino acid substitutions were introduced on 14 occasions in the main lineage (green boxes in Fig 1), while 16 different substitutions were introduced on 12 occasions in different subclades (orange boxes in Fig 1).

## Genetic diversity in the IOL

Throughout the CHIKV IOL divergence, we identified 47 mutations that occurred at different time points in viral evolution. To rationalise the impact of the mutations on viral proteins function, we mapped the emergent mutations that occurred in the IOL to solved crystal structures or ColabFold-predicted models. We analysed the replication complex and the trimeric E1-E2-E3 spike separately, as they consist of highly interactive complexes and high-quality structural models are available for these complexes. For the Nsp3, CP, and 6K proteins, which had a high number of mutations in the experimentally unsolved regions, we mapped the mutations to the ColabFold-predicted models and confirmed minimal deviation from the experimentally determined structures of their globular domains to ensure an accurate prediction. Not surprisingly, we found that most mutations occurred in the surface-exposed regions of the viral proteins and that the mutations were predominantly conservative in nature, with a few notable exceptions (Figs 2–4). We found most mutations in the E1-E2-E3 spike complex, where mutations were evenly distributed across all three proteins, and the fewest mutations in protein 6K, where only one emergent mutation was observed. An overview of all identified mutations is shown in Table 1.

## Replication complex

The replication complex consists of the proteins Nsp1, Nsp2, and Nsp4, which form a disc-like structure that docks into the neck of the ultrastructures packed with viral RNA, the so-called spherules (Fig 2A) [45,71]. In this complex, eleven monomers of the RNA capping enzyme Nsp1 form an outer ring to which the RNA-dependent RNA polymerase Nsp4 is docked. The viral protease Nsp2 also associates with the complex from the cytoplasmic side (Fig 2A) [45]. In Nsp1, two mutations, T128K and Q488R, are located on the protein surface, which probably have only minimal effects on the stability and function of the protein (Fig 2B). The conservative mutation I290V was found to be buried in the hydrophobic core of the protein and may have slight effects on protein stability, while the T376M mutation is located near the Nsp1-Nsp4 interface and thus may affect the docking of Nsp4 to the oligomeric ring of Nsp1.

Of the seven mutations found in Nsp2, three were located in the N-terminal helicase domain (S54N, H130Y and E145D) and four in the C-terminal protease domain (N495S, L539S, S566F and A793V; Fig 2C). While H130Y and E145D are conservative surface mutations that likely have limited effects on Nsp2 function, S54N is located at the interface between Nsp2 and Nsp4 so the mutation could potentially have an impact on the interaction between Nsp2 and Nsp4 and on the overall stability of the complex. Of the four mutations found in the protease domain, N495S retains a hydrophilic character and is unlikely to affect the function of the protein. The non-conservative surface mutations L539S and S566F could alter the stability of the protein or interactions with potential binding partners, and A793V introduces a larger hydrophobic moiety on the surface of the short, disordered C-terminal peptide of the Nsp2 protease domain, which is unlikely to be favourable. Being in moderate proximity to the Nsp2 active site, it is possible that the N495S, L539S and S566F mutations affect substrate recognition, as the exact substrate binding interface for CHIKV Nsp2 is not clear. Interestingly,

**Table 1. Amino acid substitutions in the IOL.**

| Protein | Amino acid | | | Conservation | Location | Effect | Node | Published functional studies |
|---------|-----|-----|-----|--------------|----------|--------|------|------------------------------|
| | Pos. | Ori. | Sub. | | | | | |
| Nsp1 | 128 | T | K | polar to charged | surface exposed | minimal | G | |
| | 290 | I | V | conservative | hydrophobic core | may effect protein stability | AA | |
| | 376 | T | M | polar to hydrophobic | ambigous | may effect docking of Nsp4, into Nsp1 ring | C | |
| | 488 | Q | R | polar to charged | surface exposed | minimal | A | |
| Nsp2 | 54 | S | N | conservative | Nsp4 interface | may impact Nsp2, Nsp4 interaction | A | |
| | 130 | H | Y | semi-conservative | surface exposed | minimal | N | |
| | 145 | E | D | conservative | surface exposed | minimal | U | |
| | 495 | N | S | conservative | surface exposed | may effect substrate recognition | Z | |
| | 539 | L | S | hydrophobic to polar | surface exposed | may alter protein stability or protein protein interactions, may effect substrate recognition | H | |
| | 566 | S | F | polar to aromatic | surface exposed | may alter protein stability or protein protein interactions, may effect substrate recognition | Y | |
| | 793 | A | V | semi-conservative | surface exposed | likely unfavorable | X | |
| Nsp3 | 59 | M | T | hydrophobic to polar | surface exposed | may have stabilizing effects | D | |
| | 217 | Y | H | semi-conservative | surface exposed | may effect RNA replication and essembly | AA | |
| | 337 | T | I | polar to hydrophobic | surface exposed | may be new interaction sites, or minimal effect | A | |
| | 338 | T | M | polar to hydrophobic | surface exposed | may be new interaction sites, or minimal effect | F | |
| | 372 | D | E | conservative | surface exposed | may be new interaction sites, or minimal effect | W | |
| | 461 | L | P | hydrophobic to cyclic | surface exposed | may be new interaction sites, or minimal effect | A | |
| | 471 | P | S | cyclic to polar | surface exposed | may be new interaction sites, or minimal effect | A | |
| Nsp4 | 55 | S | N | conservative | surface exposed | minimal | T | |
| | 75 | T | A | semi-conservative | surface exposed | minimal | A | |
| | 82 | R | S | charged to polar | surface exposed | minimal | I | |
| | 85 | R | G | charged to small flexible | surface exposed | may affect Nsp1-Nsp4 interface, may impact dimer formation | O | |
| | 254 | T | A | semi-conservative | surface exposed | minimal | A | |
| E1 | 55 | I | V | conservative | hydrophobic core | may interfere with folding | M | |
| | 136 | L | F | semi-conservative | surface exposed | minimal | S | |
| | 211 | K | E | polar to charged | surface exposed | effects adaptability to *Ae. aegypti* | L | [57–62] |
| | 226 | A | V | semi-conservative | Mxra8 interface | increases fitness in *Ae. albopictus* | B, E | [60, 63–65] |
| | 269 | M | V | conservative | surface exposed | minimal | A | |
| | 284 | D | E | conservative | surface exposed | minimal | A | |
| | 317 | I | V | conservative | surface exposed | minimal | R | |
| E2 | 74 | M | I | conservative | Mxra8 interface | minimal | S | |
| | 76 | A | T | semi-conservative | Mxra8 interface | minimal | P | |
| | 205 | G | S | small flexible to polar | surface exposed | may effect immune evasion | V | |
| | 210 | L | Q | hydrophobic to polar | surface exposed | may effect immune evasion | D | |
| | 211 | I | T | hydrophobic to polar | surface exposed | may effect immune evasion | A | |
| | 252 | K | Q | charged to polar | E3 interface | may affect E2, E3 interface | H | |

*(Continued)*

**Table 1.** (Continued)

| Protein | Amino acid | | | Conservation | Location | Effect | Node | Published functional studies |
|---|---|---|---|---|---|---|---|---|
| | Pos. | Ori. | Sub. | | | | | |
| | 264 | V | A | semi-conservative | Mxra8 interface | may affect MXRA8, spike interaction | K | [60, 66–70] |
| | 312 | T | M | polar to hydrophobic | surface exposed | minimal | A | |
| | 375 | S | T | conservative | surface exposed | minimal | A | |
| | 386 | V | A | semi-conservative | surface exposed | minimal | A | |
| E3 | 39 | V | I | conservative | E2 interface | may affect E2, E3 interface | J | |
| | 56 | P | S | cyclic to polar | surface exposed | minimal | J | |
| 6K | 8 | V | I | conservative | surface exposed | minimal | A | |
| CP | 23 | P | S | cyclic to polar | surface exposed | minimal | G | |
| | 27 | V | I | conservative | surface exposed | minimal | G | |
| | 73 | K | R | conservative | surface exposed | minimal | Z | |
| | 79 | N | S | conservative | surface exposed | minimal | Q | |

the A793V mutation reverted to A in the last common ancestor of the 2018 Thai outbreak lineage.

The Nsp4 is largely devoid of emergent mutations, with the exception of the N-terminal domain, which extends into the replication spheroid space and for which an interaction with the RNA template has been proposed [45] (Fig 2D). All of the mutations we identified in Nsp4 (S55N, T75A, R82S, R85G and T254A) are surface mutations, with R82S and R85G being the most likely to affect Nsp4 protein function due to their non-conservative nature. The R85G mutation in particular is located close to the Nsp1-Nsp4 interface and could impair the respective dimer formation.

## Spike complex

The spike complex is a trimer of E1-E2-E3 heterotrimers, which forms the icosahedral outer envelope (Fig 3A) of the virus particle and is responsible for receptor binding, membrane fusion and viral entry [72–76]. While E1 performs membrane fusion in acidic environments [77], E2 facilitates Matrix remodelling-associated protein 8 (MXRA8) receptor binding [78], and E3 protects premature exposure of the E1 fusion loop and is important for correct E1-E2 maturation [78,79]. We identified seven, ten and two emergent mutations in the E1, E2, and E3 proteins, respectively. All mutations found in E1 are surface mutations with the exception of I55V, which is located in the hydrophobic core of domain II and could interfere with the correct folding of the protein (Fig 3B). Of the remaining six mutations, three are located in the domain I and domain III regions of the protein and are located both proximal to the membrane and at the base of the spike. Mutation L136F is located on the surface of domain I, I317V is located on the surface of domain III, and mutation D284E is situated in the linker region between domains I and III. This junction region undergoes substantial rearrangement upon conversion of E1 to the fusion form [80,81], but given the conservative nature of the mutation, the rearrangement is unlikely to be affected by the identified mutation. As the surface mutations found in domains I and III are positioned at the outer surface of the spike, they could also affect the packing of neighbouring spike complexes into the icosahedral viral lattice. Of the remaining three mutations, K211E has already been described to have an impact on viral replication and the adaptability to the *Ae. aegypti* vector [57–62]. The conservative mutation M269V is located on the inner spike surface of E1 and probably has no effect on protein

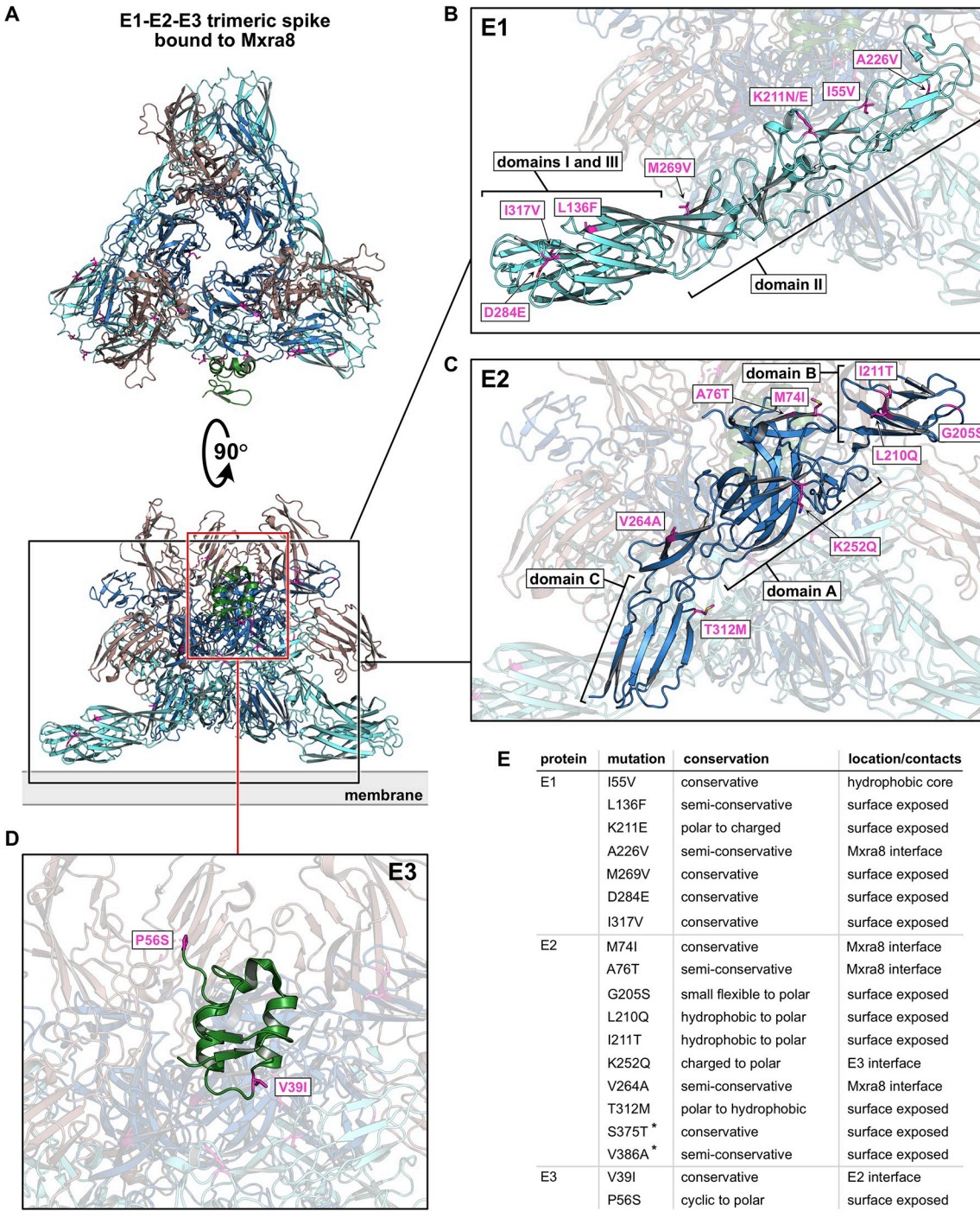

**Fig 3. Emergent mutations in the E1-E2-E3 spike complex. A)** Representative view of the Spike trimer. Structural models for visualization were obtained from PDBid 6jo8 [47]. E1 is coloured cyan, E2 blue and E3 is green. The receptor Mxra8 is included to aid the visualization of the receptor binding interface and is coloured brown. The position of the membrane at the base of the spike is indicated. All emergent mutations are shown as sticks and coloured purple as in Fig 2. **B)** Close-up view, highlighting the mutations on E1. The domains I, II and III are indicated. **C)** enlarged model of E2 highlighting emergent mutations. Domains A, B and C are indicated. **D)** Close-up view, showcasing the mutations on E3. **E)** Summary of all mutations found in the spike complex. Asterix denotes mutations that are not part of the analysed structure and are therefore not visualized.

function. The A226V mutation of E1, which is associated with increased fitness of CHIKV in *Ae. albopictus* [60,63–65], and probably contributes to the epidemic potential of CHIKV, was introduced in two different subclades at different time points (Fig 1, Node B and E). However, in the subclade following node E, two sequences, FJ000067.1 from India and FJ445428.2 from Sri Lanka, do not have valine at position 226 but alanine. In the main IOL, the ancestral alanine remained at position 226.

MXRA8 is a cell surface receptor for several arthritogenic alphaviruses, such as CHIKV [76,82], and interacts mainly with the outer crown of the spike complex consisting of three E2 proteins. Although the E2 mutations M74I and A76T are located proximal to the MXRA8 interface, they are unlikely to have a significant impact on receptor interaction given their conservative nature. In contrast, V264A is in direct contact with the MXRA8, so this mutation probably has a greater impact on the interaction between the spike complex and the cell surface receptor. Because it is exposed on the surface of the virion, E2 is also the primary target of natural and recombinant antibodies [83–85]. Several of the antibodies target the B domain of E2 [83,84] in the region where we also found three emergent mutations: G205S, L210Q, and I211T (Fig 3C). Since the same residues are targeted by antibodies, their mutation could serve as an immune evasion mechanism leading to enhanced viral fitness [83,84,86]. Interestingly, the K252Q mutation is located at the interface of E2 and E3 and is in direct contact with the V39I mutation on the E3 protein. Since these mutations do not occur in the same viral clade, they are likely not the result of coevolution, but suggest that some degree of amino acid variation is operative in this region of E1–E3 (Fig 3C). Finally, T312M is a surface mutation with likely limited effects on protein function, while S375T and V386A are located in the transmembrane region of the E2 protein with probable minimal effects on the protein function.

Apart from the V39I mutation, only one other mutation was found in the E3 protein (Fig 3D). The P56S mutation, which is located in the immediate vicinity of the C-terminal furin cleavage site [78], probably has no major influence on protein function. Both E3 mutations occur only in one clade of CHIKV with node J as the closest common ancestor (Fig 1).

## Nsp3, 6K, and capsid proteins

For the proteins that are neither part of the replication nor the spike complex, we have mapped the mutations to the individual structural models predicted by ColabFold. The Nsp3 protein, which is closely associated with the replication complex [45,87], consists of two folded globular domains, the N-terminal macro-domain and the zinc-binding domain, followed by an elongated hypervariable C-terminal disordered region (Fig 4A) [87]. The macro-domain exhibits ADP-ribosylhydrolase activity [88], while the zinc-binding domain, although poorly understood, is associated with various functions in viral genome replication and transcription that are often species- and cell type-specific [89]. The hypervariable C-terminal region has been shown to be intrinsically disordered [90] and serves as a platform for the binding of various host factors [90,91]. We identified one emergent mutation in each of the two folded domains and five mutations in the hypervariable C-terminal disordered region (Fig 4A). The M59T mutation, located on the surface of the macro-domain, is on the opposite side of the active site and might have a stabilising effect due to the transition from a hydrophobic to a hydrophilic surface-exposed amino acid. The Y217H mutation is located at the base of a small, shallow pocket on the surface of the zinc-binding domain, which could be a binding pocket. Interestingly, a reversal to an ancestral Y can be observed at this position in 2019 (Node AA, Fig 1). As the function of this domain is unclear, it is difficult to speculate on the effects of this specific mutation. However, it has been shown that the entire domain is crucial for RNA replication and viral assembly [92]. None of the five mutations found in the C-terminal domain interfere

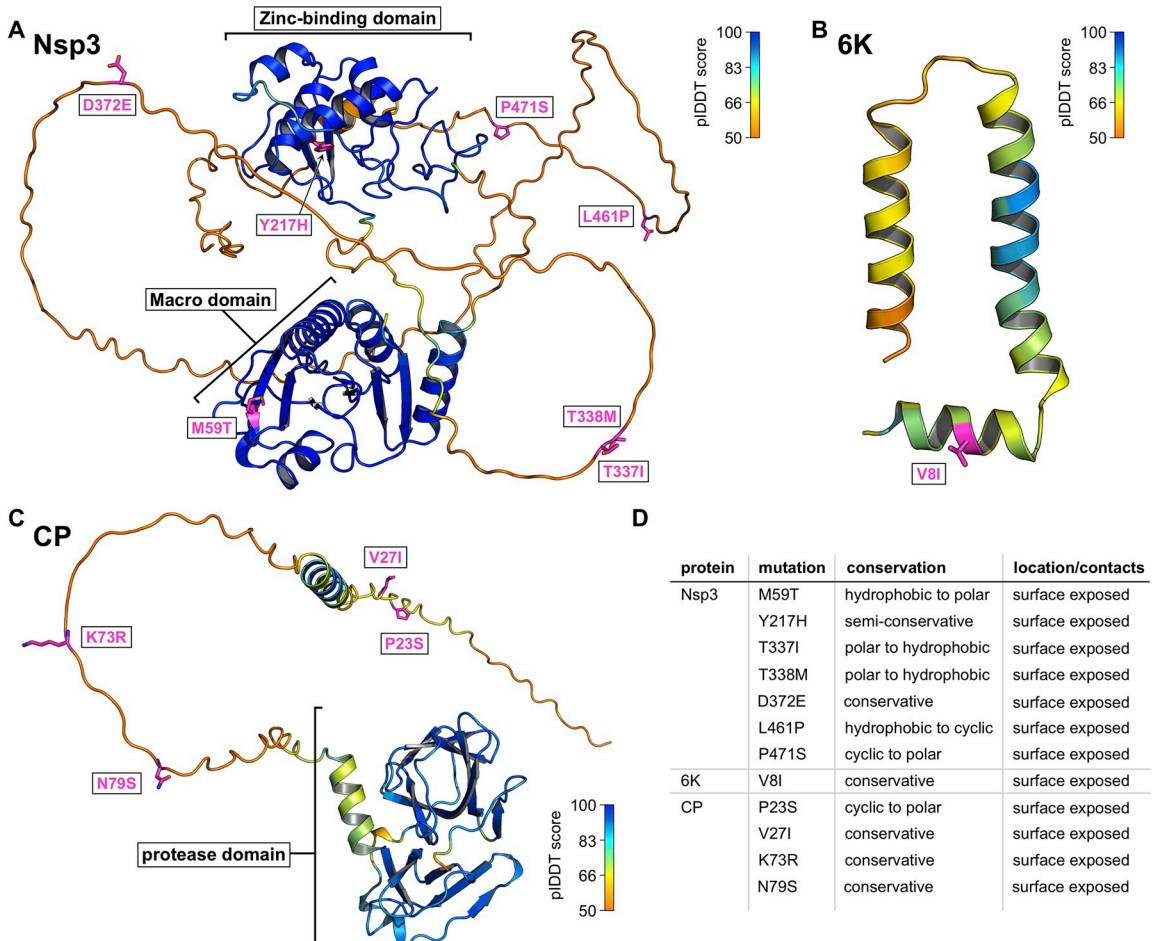

**Fig 4. Emergent mutations in the Nsp3, 6K and CP proteins.** All mutations were mapped onto the ColabFold-predicted structural models. Positions of mutations are shown as purple sticks, and the overall colouring of the proteins is according to the pLDDT score indicating the confidence of the prediction. Blue colour signifies highest prediction confidence and orange lowest as shown by the legend. **A)** Emergent mutations in Nsp3 protein. Macro and Zinc-binding domains are indicated. **B)** Predicted structure of 6K protein with highlighted V6I mutation. **C)** Capsid protein model with highlighted mutations in the disordered N-terminal tail of the protein. **D)** Summary of all emergent mutations found in the Nsp3, 6K and capsid proteins.

with the previously described short linear motifs found in this region, which interact with amphiphysin-SH3 (which is recruited by the virus to promote viral RNA replication) [93] and the G3BP-NTF2 domains (which are hijacked to block stress granule formation) [94]. Given the high density of short linear motifs in the disordered regions of viral proteins and the fact that the C-terminal disorder domain of Nsp3 acts as an interaction hub for host factors, it is possible that the T337I, T338M, D372E, L461P, and P471S mutations create new binding sites for host proteins or destroy existing ones [95]. Alternatively, these mutations might not have significant effects on the function of the protein and might be consequences of random drift.

The 6K protein is a poorly understood, highly hydrophobic protein that forms hexameric ion channels in the endoplasmic reticulum (ER) membrane [96,97]. We have identified only one conservative mutation, V8I, which likely has limited or no effect on protein function (Fig 4B).

Finally, the capsid protein (CP) is a multifunctional protein with an N-terminal, positively charged, intrinsically disordered region involved in RNA encapsidation [98], and a C-terminal, chymotrypsin-like protease domain that binds to the transmembrane helix of E2 and forms the inner lattice of the mature viral nucleocapsid [79,99]. All four mutations found in

the CP protein, P23S, V27I, K73R, and N79S, are located in the N-terminal disordered region (Fig 4C). The mainly conservative nature makes it unlikely that these mutations have a major impact on the function of CP. This, and the fact that we found no emergent mutations in the C-terminal protease domain, suggests that the CP protein is under tight evolutionary constraints that allows only very limited variation in the amino acid sequence.

We speculated about the effect of the mutations based on their position in the structure, but further experimental validations should be performed to accurately determine the effect of individual mutations. This opens an interesting avenue for future work.

## Discussion

CHIKV has left an indelible mark on the global landscape of infectious diseases, and its emergence and spread over time provide valuable insights into the complexity of vector-borne diseases. The current diversity of CHIKV is thought to have originated in sub-Saharan Africa in the 1920s [6]. This is consistent with our analysis, where the deepest split in our CHIKV tree is estimated to be in the 1950s. The emergence of the Asian genotype shortly thereafter in the 1930s marked the beginning of a series of events that eventually led to the formation of distinct lineages. These genotypes, including the West African, the Asian, and the ECSA genotype with the IOL, illustrate the intricate evolutionary history of CHIKV. Each genotype has played a unique role in the global spread of the virus.

Our analysis shows that there is considerable interregional transmission of CHIKV. The Southeast Asian region, particularly India, stands out as an important interregional transmission site linking CHIKV isolates from other regions. Sub-Saharan Africa, where CHIKV first emerged, also plays a central role in the global spread, albeit to a lesser extent than Southeast Asia. South America appears to be the primary source of intra-continental spread, rather than a source of transmission of CHIKV to other continents. This long-distance transmission emphasises the adaptability of CHIKV to different ecological and environmental conditions, as it can be transmitted in both urban and sylvatic cycles [100]. This provides evidence of the resilience and adaptability of CHIKV as it navigates different regions and ecosystems. The possibility of transmission in both urban and sylvatic environments give CHIKV the opportunity to spread through infected human travellers and cause new outbreaks, but also to circulate locally and establish an endemic occurrence of the virus.

The IOL represents an intriguing aspect of the global spread of CHIKV. It shows a multitude of bidirectional transmission events linking Southeast Asia, India, East Africa, the Arabian Peninsula, and Europe. This lineage emphasises the intricate network of CHIKV transmission in the Indian Ocean region and highlights the role of different regions in maintaining the presence of the virus.

Phylogenetic analysis of CHIKV Thai strains isolated between 2018 and 2020 during the large outbreak in Thailand revealed that they are mapped within the IOL to the ECSA genotype, the same genotype responsible for the massive Thailand outbreak in 2008–2009. The strains from the 2008–2009 outbreak however, possess the E1 A226V mutation, which is associated with enhanced transmission by *Ae. albopictus*, compared to strains circulating before 2008 [53,60,64,65,67,101,102]. A 2021 study by Khongwichit et al. found that none of the ECSA strains isolated during the second massive outbreak in Thailand from late 2018 to early 2020 carried this E1 A226V mutation [24], nor did we find it in our Thai isolates (Fig 1). Instead, the new Thai strains had the ancestral alanine at position 226 of the E1 envelope glycoprotein, showing similarities to previous outbreaks in Thailand in 1958 [23]. This leads us to hypothesise that there must have been other factors in the 2018 outbreak that lead to the rapid spread of the virus.

We and others found that the 2018–2020 Thai strains had additional mutations of interest, such as E1 K211E (Fig 1, Node L and Fig 3B) and E2 V264A (Fig 1, Node K and Fig 3C) [24]. It has been reported that positive selection had a dramatic effect on the alteration of the amino acid residue from lysine (K) to glutamic acid (E) at position 221 of the E1 protein and that mutations on the E1 and E2 envelope glycoproteins in general can affect the vector competence, transmission efficiency, and pathogenicity of the virus [60,66–70]. The V264A substitution is located at the MXRA8 receptor-binding interface and the mutation could alter the interaction between the viral spike complex and the cell surface receptor. The E1 K211E mutation has been associated with enhanced viral infection in *Ae. aegypti* and has also been reported in other regions [103–107]. This adaptation to a different vector may have influenced the increased spread of CHIKV in Thailand in 2018–2019. Consistently, all sequences isolated and sequenced in this study contain alanine at position 226 of the E1 protein and carry the mutations E1 K211E and E2 V264A.

Two other notable mutations in the structural protein E2 are I211T (Fig 1, Node A and Fig 3C) and G205S (Fig 1, Node V and Fig 3C). The I211T mutation occurs at the IOL ancestral node while the G205S substitution occurs at node V, probably in early 2016, shortly before the progenitor of the 2018 Thai outbreak began to circulate in Bangladesh. Both mutations are located in the region that has been described as critical for antibody binding and recognition [83,84]. Mutations at these positions could therefore lead to evasion of the immune system, increased spread, and higher virulence in the population.

Another substitution that occurred later, around the end of 2017, in Bangladesh is the Nsp2 V793A reverse mutation (Fig 1, Node X and Fig 2C). As this mutation is located at the very end of the C-terminal disordered tail of Nsp2 (as predicted by ColabFold), which was not resolved in the crystallisation studies [46], we propose that the introduction of the hydrophobic moiety interferes with the optimal function of Nsp2, leading to the observed V793A back mutation. Importantly, the alanine at position 793 is also present in the rapidly expanding ECSA 2 lineage in South America, suggesting that it may have a beneficial effect on viral transmission (Fig 1).

The comparison of the phylogenetic relationship of the CHIKV sequences from the 2018–2020 Thai outbreaks with other global sequences showed that the most recent outbreak in Thailand did not originate from the strain circulating in the country. It also belongs to the IOL, but probably originated from other countries in South Asia, most likely from Bangladesh via Myanmar in late 2017 or early 2018. The timing suggests a gradual overland introduction into Thailand from Bangladesh via Myanmar, e.g. through travelling and resettlement of people with subsequent spread within Thailand and spill over to China and Cambodia in mid-2019 and 2020. Due to the political situation in Myanmar, no information could be obtained on the number of positive CHIKV cases in the years between 2016 and 2018.

The emergence of CHIKV outbreaks in certain regions, such as the outbreak in Thailand in 2018, is an example of the ability of the virus to re-emerge and spread rapidly. This emphasises the importance of monitoring and understanding the dynamics of CHIKV transmission to take effective public health measures. It is not yet fully understood whether the re-emergence of the virus is caused by purely urban cycles with occasional re-introduction from other countries or whether there also is a sylvatic component. CHIKV could circulate in a sylvatic cycle of non-human primates and mosquitoes and remain undetected in the wild until a spill over event into the urban human mosquito cycle occurs, which in some cases could cause new local outbreaks [10,11]. Both scenarios are possible and plausible. In some cases, very low genetic variation can be detected in strains occurring in the same geographical area years apart (Fig 1, observed in Brazil in ECSA 2 or after Node M in India). In these cases, undetected sylvatic transmission could be suggested as a silent reservoir. However, in cases such as the 2018 Thai

outbreak, introduction from an urban cycle in a neighbouring country is more likely considering the genetic relatedness and timing of virus spread.

A global increase in CHIKV circulation was detected in 2023. By 30 November, more than 460 000 cases had been reported accompanied by 360 deaths [108]. South America was particularly affected. Argentina and Uruguay reported local transmission for the first time in 2023 [109]. Contributing factors include climate change, which leads to changes in vector activity and distribution, and increased human travel, which plays an important role in the spread of CHIKV in South America and globally [110–114]. Unusual temperature spikes, prolonged warm spells, and altered rainfall patterns combined with increased humidity have created conditions that favour the survival and proliferation of *Ae. aegypti* and *Ae. albopictus* mosquitoes in regions where they were previously absent [115–120]. The emergence of CHIKV in the Caribbean islands, a favourite destination for tourists from North America and Europe, creates additional new opportunities for intercontinental transmission of the infection [114]. Unanticipated and rapid urbanization further promotes the spread of the virus, as *Ae. aegypti* and *Ae. albopictus* are particularly attracted to urban areas and warm environments. These mosquitoes utilise water-containers in or near households, such as plant pots and vases, for breeding, which further increases the rate of transmission [121,122].

In summary, CHIKV has a rich evolutionary history, originating in sub-Saharan Africa and spreading worldwide through complex genotypes, lineages, and transmission centres. A detailed analysis of Thai strains from 2018 to 2020 shows that unique mutations associated with virus replication, receptor binding and transmission occur throughout the genome, suggesting alternative factors for the rapid spread in the 2018 outbreak. The ability of the virus to re-emerge and spread rapidly, combined with climate change and urbanisation, poses an ongoing public health challenge. Monitoring and understanding CHIKV dynamics remain critical to an effective response to the unpredictable outbreaks of the virus.

## Supporting information

**S1 Fig. A maximum likelihood tree of all 2,564 complete or near complete CHIKV genome sequences.** Sequences that were included in the subsampled dataset were colour coded according colours in Fig 1.
(PDF)

**S2 Fig. TempEst regression for the ECSA genotype.** Data points were colour coded according colours in Fig 1.
(PDF)

**S3 Fig. Maximum clade credibility tree from Fig 1, here depicting all branches with posterior probability of ≥ 0.95.**
(PDF)

**S1 Table. Includes strain name, sampling date, geographical location and NCBI GenBank accession number for all twelve patient samples sequenced in the present study.**
(XLSX)

## Author Contributions

**Conceptualization:** Xavier de Lamballerie, Weifeng Shi, John H.-O. Pettersson.

**Formal analysis:** Michael W. Gaunt, Jon Bohlin, Jenny C. Hesson, Cixiu Li.

**Funding acquisition:** Åke Lundkvist, Weifeng Shi, John H.-O. Pettersson.

**Investigation:** Janina Krambrich, Filip Mihalič.

**Visualization:** Janina Krambrich, Filip Mihalič.

**Writing – original draft:** Janina Krambrich, Filip Mihalič.

**Writing – review & editing:** Janina Krambrich, Michael W. Gaunt, Jon Bohlin, Jenny C. Hesson, Åke Lundkvist, Xavier de Lamballerie, Weifeng Shi, John H.-O. Pettersson.

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
