## [Decision Letter · Decision Letter 0]

30 May 2024

Dear Pettersson,

Thank you very much for submitting your manuscript "The evolutionary and molecular history of a chikungunya virus outbreak lineage" for consideration at PLOS Neglected Tropical Diseases. As with all papers reviewed by the journal, your manuscript was reviewed by members of the editorial board and by several independent reviewers. The reviewers appreciated the attention to an important topic. Based on the reviews, we are likely to accept this manuscript for publication, providing that you modify the manuscript according to the review recommendations. 

Sincerely,

Wojciech Makalowski

Guest Editor

Andrea Marzi

Section Editor

Reviewer's Responses to Questions

**Key Review Criteria Required for Acceptance?**

**Methods**

-Are the objectives of the study clearly articulated with a clear testable hypothesis stated?

-Is the study design appropriate to address the stated objectives?

-Is the population clearly described and appropriate for the hypothesis being tested?

-Is the sample size sufficient to ensure adequate power to address the hypothesis being tested?

-Were correct statistical analysis used to support conclusions?

-Are there concerns about ethical or regulatory requirements being met?

Reviewer #1: The study is designed for multi-year phylogenetic analysis of CHIKV using available public data The authors sequenced 12 additional CHIKV sequence from travelers returning from Thailand. The number of samples used for the analysis is sufficient. Conclusions are very well supported by the analysis and statistical analysis employed.

Reviewer #2: yes

**Results**

-Does the analysis presented match the analysis plan?

-Are the results clearly and completely presented?

-Are the figures (Tables, Images) of sufficient quality for clarity?

Reviewer #1: The analysis matches with the plan to describe the evolutionary of CHIKV outbreak. Figures and results are also not that clear (see below).

Reviewer #2: yes

**Conclusions**

-Are the conclusions supported by the data presented?

-Are the limitations of analysis clearly described?

-Do the authors discuss how these data can be helpful to advance our understanding of the topic under study?

-Is public health relevance addressed?

Reviewer #1: Conclusions seem to be supported by the data. The authors, however, did not mention any limitations of the analysis.

Reviewer #2: yes

**Editorial and Data Presentation Modifications?**

Reviewer #1: There are some minor revisions to improve the readability.

1. Please write in more detail how the sequences were assembled.

2. Please separate Figure 1 into Figure 1A for the phylogenetic tree and 1B for the map of the movement.

3. There seems to be detachment of the map of the virus movement and its explanation in the manuscript. For example, line 212 says, “islands (Madagascar, Mauritius, Mayotte, Comoros) in the 1960s and to India in the 1980s (blue)”, however the map shows that the year of the spread to India is 2003. Also, line 210 says, “Central Africa to the Americas (red) in the late 1980s and early 2010s (1995 in the USA and 2014”, however the map shows that the spread to the USA in 1979 and to Brazil in 2013.

4. The map in Figure 1 shows red arrow pointing from Cameroon to Madagascar. Is it not blue arrow? Also, the arrow pointing the movement of ECSA2 to ECSA1 from Uganda to Senegal in 1963 does not seem to fit the phylogenetic tree.

Reviewer #2: Abstract:

1) “…we also identified amino acid substitutions that may be associated 14 with immune evasion, increased spread, and virulence.“ and the next sentence “…which are highly relevant as they may 16 lead to changes in vector competence, transmission efficiency and pathogenicity of the virus” are duplicates. Rephrase, please.

2) I would prefer to have both latin names spelled full at the first encountering in the text: Aedes aegypti and Aedes albopictus (vs. Ae. albopictus) line 38

3) A badly phrased sentence – line 42-43

4) Line 91 – “mutational outbreak lineage analysis” sounds strange to me, but I am not an epidemiologists.

5) Line 92 – time estimation for what? Please, specify better.

Methods:

1) 12 positive patients out of? I am rather curious how relevant is the study in Europe.

2) Line 114-115 – genomes generated above? Do author mean the consensus sequences? It can be explained a bit clearer.

3) Line 123-124 – where the older datasets were taken from? Ref to the database or a source should be provided (years 1953-2023)

4) Line 137 – how the mutations were mapped on the models?

Main text:

1) Line 235, line 268 – typos in the sentences?

Supplementary:

I suggest to make the big tree of the Sup. Fig.1 circular. When rectangular, it is not possible to read.

**Summary and General Comments**

Reviewer #1: The manuscript by Krambich J et al. titled The Evolutionary and Molecular History of a Chikungunya Virus Outbreak Lineage is interesting for general reader. Using sequence of the CHIKV from travelers to Thailand, they are able to make a connection how the virus lineage is connected to the previous outbreaks and also able to give the reader a picture of the spread of the virus, especially the ECSA genotype. Therefore, this is a very nice paper and is good for publication after addressing the minor revisions.

Reviewer #2: The authors did a detailed work on tracing the spread history of the chikungunya virus around the globe. This is an important aspect of epidemiology and virology and the study helps our understanding of these processes for CHIKV and for other viruses, in general. Moreover, the analyses of the mutations and their possible consequences on the virus life cycle are a meticulous and well done work. This is also a great contribution to molecular biology and evolution of the mechanisms of viral infections. I recommend this work to publishing with some minor changes. See below.

PLOS authors have the option to publish the peer review history of their article (what does this mean?). If published, this will include your full peer review and any attached files.

Reviewer #1: No

Reviewer #2: No

Figure Files:

Data Requirements:

Reproducibility:

References

---

## [Editor Report · Decision Letter 1]

8 Jul 2024

Dear Pettersson,

We are pleased to inform you that your manuscript 'The evolutionary and molecular history of a chikungunya virus outbreak lineage' has been provisionally accepted for publication in PLOS Neglected Tropical Diseases.

Best regards,

Wojciech Makalowski

Guest Editor

Andrea Marzi

Section Editor

---

## [Editor Report · Acceptance letter]

22 Jul 2024

Dear Pettersson,

We are delighted to inform you that your manuscript, "The evolutionary and molecular history of a chikungunya virus outbreak lineage," has been formally accepted for publication in PLOS Neglected Tropical Diseases.

Best regards,

Shaden Kamhawi

co-Editor-in-Chief

Paul Brindley

co-Editor-in-Chief
